# Evolving Treatment Options for Metastatic Renal Cell Carcinoma (mRCC)

Eun-mi Yu [1,†], Mythri Mudireddy [2,†], Ishan Patel [2] and Jeanny B. Aragon-Ching [1,*]

1   GU Medical Oncology, Inova Schar Cancer Institute, Fairfax, VA 22031, USA
2   Department of Hematology and Oncology, Inova Schar Cancer Institute, Fairfax, VA 22031, USA
*   Correspondence: jeanny.aragon-ching@inova.org; Tel.: +571-472-4724
†   These authors contributed equally to this work.

**Abstract:** Approximately a third of patients diagnosed with kidney cancer in the United States present with advanced disease and those who present with distant metastases historically had dismal 5-year relative survival. However, over the last several years, advancements have led to improved life expectancy and patient outcomes in those who develop advanced renal cell carcinoma. Metastatic clear cell renal cell carcinoma (mccRCC) treatment has rapidly evolved with multiple drug approvals since 2006. Moreover, multiple combination regimens including a vascular endothelial growth factor tyrosine kinase inhibitor (VEGF-TKI) plus immune checkpoint inhibitor (ICI) and the combination of ipilimumab plus nivolumab have supplanted first-line VEGF-TKI monotherapy. Thus, the insights we gained from prospective randomized controlled trials focusing on systemic therapy beyond first-line therapy in mRCC patients treated in the TKI monotherapy era quickly became less relevant with the adoption of contemporary first-line combination regimens. Herein, we will review contemporary first- and second-line therapies for mccRCC, as well as highly anticipated clinical trials looking into novel regimens beyond first-line therapy in patients who have received combination therapy.

**Keywords:** metastatic renal cell carcinoma; checkpoint inhibitors; VEGF inhibitors; nivolumab; ipilimumab

## 1. Introduction

Renal cell cancer afflicted about 79,000 patients in 2022 and was responsible for deaths of about 13,920 patients in the United States [1]. About one-third of patients in the United States diagnosed with kidney cancer have advanced disease at the time of diagnosis, and those with distant metastases historically had a low 5-year relative survival rate of only 15%. Clear cell renal cell carcinoma comprises the most common histology for all renal cell carcinomas [2,3], and is the main focus of this review. While surgery has been the primary treatment for early-stage, resectable renal cell carcinomas (RCCs), the treatment of metastatic clear cell retablenal cell carcinoma has evolved from interleukin therapy to VEGF-TKIs over the years, and in recent times to the use of ICIs or immuno-oncology (IO) therapies. In addition, the evolution of combination regimens (IO/VEGF-TKIs or IO/IO) as first-line treatment has improved survival outcomes in these patients and frequent discussion regarding toxicities, sequencing, and treatment resistance often abounds. Choice of therapy hinges upon multiple factors although the use of a prognostic model with the International Metastatic Renal-Cell Carcinoma Database Consortium (IMDC) criteria is still the basis for most contemporary clinical trial risk stratification and treatment assignment [4]. This review discusses the evolution of treatment from front-line therapy to second-line treatment in specific timelines (see Figure 1) not just in the VEGF-TKI era but most especially in the contemporary management of mccRCC after IO exposure.

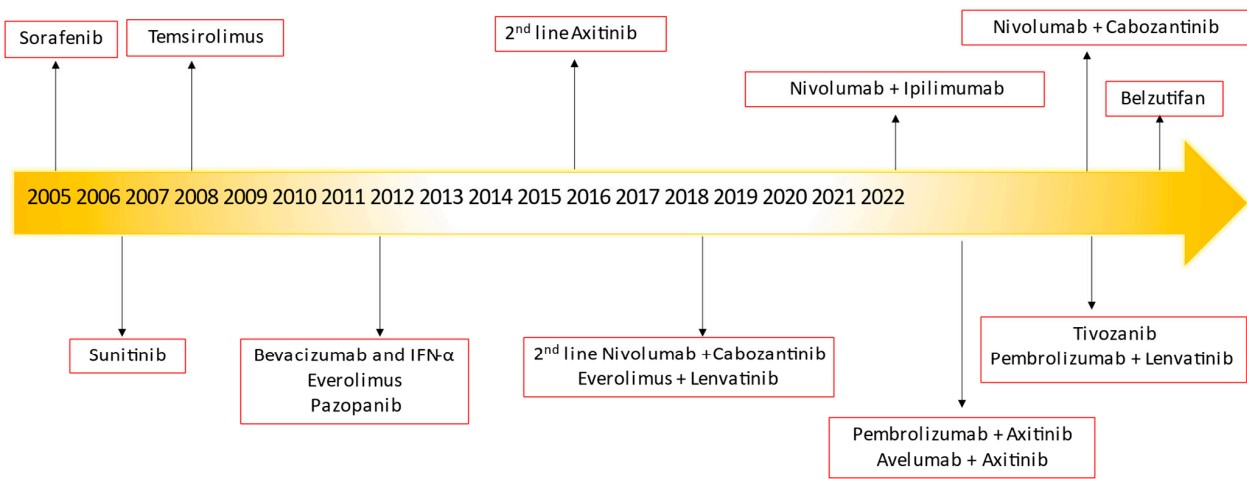

**Figure 1.** Evolving landscape of metastatic renal cell carcinoma treatment.

## 2. Front-Line Therapy for mRCC

Based on the landmark trials utilizing sunitinib and pazopanib, VEGF-TKI monotherapy was established as the first-line therapy of choice for newly diagnosed mccRCC during the mid-2000s. However, combination therapy with IO agents has revolutionized the standard-of-care and first-line treatment options, which will be discussed herein (Table 1).

**Table 1.** Select key studies on first-line treatment with metastatic RCC.

| NCT# (Phase) | N | Title | Primary Endpoint(s) | Arms | Results |
|---|---|---|---|---|---|
| NCT03937219 (Ph 3) | 855 | Study of cabozantinib in combination with nivolumab and ipilimumab in patients with previously untreated advanced or metastatic renal cell carcinoma (COSMIC 313) | PFS | Nivolumab + ipilimumab + cabozantinib versus nivolumab + ipilimumab | NR vs. 11.3 months |
| NCT02231749 (Ph 3) | 1096 | Nivolumab plus ipilimumab versus sunitinib and advanced renal cell carcinoma (CheckMate 214) | OS, PFS, ORR in intermediate- and poor-risk patients | Nivolumab plus ipilimumab followed by nivolumab versus sunitinib | 18-month OS, 75% vs. 60%; mOS, NR vs. 26 months; ORR, 42% vs. 27%; PFS 11.6 months vs. 8.4 months |
| NCT02684006 (Ph 3) | 886 | Avelumab plus axitinib versus sunitinib for advanced renal cell carcinoma (JAVELIN Renal 101) | PFS OS in PD-L1-positive tumors | Avelumab + axitinib versus sunitinib | PFS, 13.8 months vs. 7.2 months; OS not statistically significant |

**Table 1.** *Cont.*

| NCT# (Phase) | N | Title | Primary Endpoint(s) | Arms | Results |
|---|---|---|---|---|---|
| NCT03141177 (Ph 3) | 651 | Nivolumab plus cabozantinib versus sunitinib for advanced renal cell carcinoma (CheckMate 9ER) | PFS | Nivolumab plus cabozantinib versus sunitinib | 16.6 months vs. 8.3 months |
| NCT02853331 (Ph 3) | 840 | Pembrolizumab plus axitinib versus sunitinib for advanced renal cell carcinoma (KEYNOTE-426) | PFS OS | Pembrolizumab plus axitinib versus sunitinib | 12-month OS, 89.9% vs. 78.3%; PFS, 15.1 months vs. 11.1 months |
| NCT028211861 (Ph 3) | 1069 | Lenvatinib plus pembrolizumab or everolimus for advanced renal cell carcinoma (CLEAR trial) | PFS | Lenvatinib plus pembrolizumab versus sunitinib, lenvatinib plus everolimus versus sunitinib | 23.9 months versus 9.2 months 14.7 months versus 9.2 months |

PFS = progression-free survival; OS = overall survival; NR = not reached.

The role of dual checkpoint inhibition was initially tested using a combination of nivolumab and ipilimumab (*n* = 550) which was compared to sunitinib (*n* = 546) in the CheckMate 214 trial which was a phase III randomized controlled trial (RCT) [5,6], which served to investigate endpoints that included objective response rate (ORR), progression-free survival (PFS), and overall survival (OS) among IMDC intermediate- and poor-risk mccRCC patients, which was the intention to treat (ITT) population of patients. After a median follow-up of 25.2 months, the combination of nivolumab and ipilimumab showed a higher ORR and complete response rate compared to sunitinib, at 42% and 9% versus 27% and 1%, respectively. The nivolumab/ipilimumab arm also showed a higher median PFS of 11.6 months compared to 8.4 months with sunitinib [5]. After 42 months of follow-up, the nivolumab/ipilimumab arms showed better OS and PFS in the intermediate-risk and poor-risk patients, with higher ORR at 42.1% compared to 26.3% in the sunitinib arm. However, inferior ORR was observed in favorable-risk patients treated with nivolumab/ipilimumab at 28.8% compared to 54% for sunitinib. The hazard ratio (HR) for death was 1.19 (95% CI, 0.77–1.85) in this patient population [6]. In addition, use of high-dose corticosteroids, defined as a dosage of prednisone of 40 mg or higher, was encountered in 157 of 547 patients (28.7%) of those treated with nivolumab/ipilimumab to manage any-grade treatment-related adverse events (TRAEs). Regardless, this trial led to the first dual checkpoint inhibitor therapy combination (also referred to as IO/IO therapy) with FDA approval for poor- and intermediate-risk patients, aligning with the intention to treat (ITT) analysis population corresponding to CheckMate 214. Inferior outcomes were noted in the IMDC favorable-risk patients treated with IO/IO compared to sunitinib. Hence, FDA approval is limited to intermediate-risk and poor-risk IMDC group patients and reflected in national guidelines.

Given the success of IO/IO combination therapy and known benefits of VEGF-TKI monotherapy, there came about increased interest in combined IO/VEGF-TKI regimens. Several studies were initiated using a combination approach. A phase Ib trial investigated the role of a combination of a monoclonal programmed cell death protein 1 (PD-1) antibody, pembrolizumab, and a VEGF-TKI, axitinib, with an ORR of 73% in advanced RCC patients [7]. As a result, a multicenter, international, open-label phase III RCT, KEYNOTE-426, was conducted to compare the efficacy of pembrolizumab plus axitinib in 432 patients versus sunitinib in 429 patients [8]. The trial was designed to evaluate primary endpoints of PFS and OS in the ITT population, with objective response rate (ORR) as a key secondary endpoint. The first interim analysis of the trial with a median follow-up period of 12.8 months revealed that patients receiving treatment with pembrolizumab and axitinib

had a higher one-year survival rate compared to those in the sunitinib arm (89.9% vs. 78.3%), with no median survival time reached in either arm. The treatment with pembrolizumab and axitinib also showed a lower risk of death than sunitinib (HR 0.53, 95% CI 0.38–0.74, $p < 0.0001$). Furthermore, the pembrolizumab and axitinib group exhibited an improved objective response rate (59.3% vs. 35.7%) and progression-free survival (15.1 months vs. 11.1 months; HR 0.69, 95% CI 0.57–0.84, $p < 0.001$) compared to sunitinib. Notably, patients in the pembrolizumab and axitinib arm demonstrated improved OS and progression-free survival regardless of PD-L1 expression or IMDC risk category. The incidence of grade 3 or higher adverse events of any cause was higher in the pembrolizumab plus axitinib arm (75.8%) than in the sunitinib arm (70.6%). In an updated analysis of the KEYNOTE-426 clinical trial with a longer follow-up period of 30.6 months, improved OS was demonstrated (median survival not reached vs. 35.7 months; HR 0.68, 95% CI 0.55–0.85, $p = 0.0003$) and PFS (median PFS 15.4 months vs. 11.1 months; HR 0.71, 95% CI 0.60–0.84, $p < 0.0001$), respectively, in the pembrolizumab and axitinib arm versus sunitinib arm [9].

Another combination of IO/VEGF-TKI was studied with avelumab and axitinib in the early phase Ib, open-label JAVELIN Renal 100 trial with promising results including a 78% disease control rate and 58% ORR [10]. This trial led to the multicenter, randomized, open-label, phase III trial JAVELIN Renal 101 which compared the combination of avelumab and axitinib ($n = 442$) with standard-of-care sunitinib ($n = 444$) with co-primary endpoints of OS and PFS in patients with PD-L1-positive tumors ($n = 560$) and secondary endpoints of PFS and OS in the overall population regardless of PD-L1 expression [11]. In the first interim analysis, the avelumab and axitinib arm demonstrated a median PFS of 13.8 months in patients with PD-L1-positive tumors compared to 7.2 months in the sunitinib arm (HR 0.61, 95% CI 0.47–0.79, $p < 0.001$). However, there was no statistically significant difference in OS between the two groups with 37 deaths in the combination arm versus 44 deaths in the sunitinib arm (HR 0.82, 95% CI 0.53–1.28, $p = 0.38$). In the overall population, the avelumab and axitinib arm had an improved PFS of 13.8 months compared to 8.4 months in the sunitinib arm (HR 0.69, 95% CI 0.56–0.84, $p < 0.001$). Nonetheless, this was not the primary endpoint, and the OS data were immature (HR 0.78, 95% CI 0.55–1.08, $p = 0.14$). In the updated efficacy data after the second interim analysis, with a median follow-up of 13 months, the avelumab and axitinib arm showed improved PFS in patients with PD-L1-positive tumors (median PFS 13.8 months vs. 7.0 months; HR 0.62, 95% CI 0.490–0.777, one-sided $p < 0.0001$) and in the overall population (median PFS 13.3 months vs. 8.0 months; HR 0.69, 95% CI 0.574–0.825, one-sided $p < 0.0001$) without OS benefit [12]. Regardless, the combination of avelumab and axitinib was approved by the FDA in May 2019 for the first-line treatment of advanced clear cell RCC patients.

A phase III randomized controlled trial named CheckMate 9ER compared cabozantinib and nivolumab combination therapy ($n = 323$) to sunitinib ($n = 328$) for the treatment of metastatic clear cell renal cell carcinoma. The trial was open-label and conducted across multiple countries [13]. The primary endpoint was PFS, while OS, ORR, and safety were all secondary endpoints. The results after a median follow-up of 32.9 months indicated that the investigational arm had a longer median overall survival than the sunitinib arm (37.7 vs. 34.3 months; HR 0.70, 95% CI 0.55–0.90). Similarly, the investigational arm had a longer median PFS (16.6 vs. 8.3 months; HR 0.56, 95% CI 0.46–0.68) and higher ORR (55.7%, 95% CI 50.1–61.2 vs. 28.4%, 95% CI 23.5–33.6). Grade 3 TRAEs were more frequent in the combination arm compared to sunitinib (65.0% vs. 54.1%). In January 2021, the FDA approved the combination of cabozantinib plus nivolumab for first-line treatment of mccRCC for all IMDC risk groups.

The CLEAR study was a more contemporary three-arm, phase III trial that investigated the efficacy of lenvatinib combined with either everolimus ($n = 357$) or pembrolizumab ($n = 355$) as a first-line treatment for patients with advanced clear cell renal cell carcinoma [14]. The trial compared these combinations to sunitinib ($n = 357$), with PFS as the primary endpoint and OS, ORR, and safety as secondary endpoints. Following a median follow-up of 26.6 months, the results showed that the two investigational arms had a

significantly longer PFS than the sunitinib arm, with the lenvatinib plus pembrolizumab arm having a PFS of 23.9 months and the lenvatinib plus everolimus arm having a PFS of 14.7 months, compared to 9.2 months in the sunitinib arm. The HR for PFS was 0.39 (95% CI 0.32–0.49; $p < 0.001$) in the lenvatinib plus pembrolizumab arm and 0.65 (95% CI, 0.53–0.80; $p < 0.001$) in the lenvatinib plus everolimus arm. In terms of OS, no median OS was reached in any arm. However, the lenvatinib plus pembrolizumab arm had a significantly longer OS than the sunitinib arm (HR 0.66, 95% CI 0.49 to 0.88; $p = 0.005$), while no OS benefit was observed in the lenvatinib plus everolimus arm compared to the sunitinib arm. The ORR was also higher in the lenvatinib plus pembrolizumab arm (71%) and lenvatinib plus everolimus arm (53.5%) than the sunitinib arm (36.1%). Adverse events of any grade were similar in the two investigational arms compared to sunitinib (99.7% vs. 98.5%). Grade 3 or higher adverse toxicity occurred in 82.4% vs. 83.1% vs. 71.8 % of the patients who received lenvatinib plus pembrolizumab vs. lenvatinib plus everolimus vs. sunitinib, respectively. Based on the CLEAR study data, the FDA approved the combination of lenvatinib plus pembrolizumab for the first-line treatment of patients with advanced clear cell RCC in August 2021.

In the CABOSUN trial conducted by ALLIANCE, which was a phase II clinical trial, cabozantinib monotherapy ($n = 79$) was compared to sunitinib ($n = 78$), a VEGF-TKI monotherapy. The trial's primary endpoint was PFS, and secondary endpoints included OS, ORR, and safety. The cabozantinib arm showed a longer PFS of 8.6 months compared to the sunitinib arm's PFS of 5.3 months (HR 0.48, 95% CI 0.31–0.74; $p = 0.0008$). The ORR was 20% for cabozantinib and 9% for sunitinib. The incidence rate of grade 3 or higher toxicity was increased in the cabozantinib arm at 68% compared to the sunitinib arm at 65%. However, there was no statistically significant difference in median OS observed between the cabozantinib arm at 26.6 months and the sunitinib arm at 21.2 months (HR 0.80, 95% CI 0.53–1.21). Despite this, cabozantinib monotherapy was approved by the FDA in December 2017 as a first-line treatment option for advanced clear cell RCC, regardless of any IMDC risk group, which differs from the population studied in the CABOSUN trial [15].

Another combination which has since been termed the "triplet" therapy trial, COSMIC 313, was a phase III study conducted across multiple countries and designed to evaluate the effectiveness of triplet therapy in the treatment of mRCC [16]. The trial results were presented at the European Society for Medical Oncology (ESMO) Congress 2022. The study compared the combination of cabozantinib, ipilimumab, and nivolumab ($n = 428$) with nivolumab, ipilimumab, and placebo ($n = 427$). The primary endpoint of the study was PFS, which was measured in the ITT population of the first 550 randomized patients. The assessment was carried out by blinded independent radiology review per RECIST 1.1 criteria. The median PFS for the combination of cabozantinib, nivolumab, and ipilimumab was not reached (95% CI: 14.0–not estimable), whereas for the combination of nivolumab and ipilimumab, the median PFS was 11.3 months (95% CI: 7.7–18.2). The ORR was 43% (95% CI: 37.2–49.2) and 36% (95% CI: 30.1–41.8) for the triplet and doublet arms, respectively. The incidence of TRAEs was higher in the triplet arm (73%) than in the doublet arm (46%), with discontinuation rates due to TRAEs of 12% and 5%, respectively. Although these data have not yet led to FDA approval, they provide evidence for the use of triplet therapy as a first-line treatment option for mRCC.

## 3. Second-Line Treatment for mRCC

Despite significant advancements in the primary treatment alternatives for metastatic RCC, there remain challenges in managing progressive disease. Multiple historic second-line therapeutic options were based on progression after first-line VEGF-TKI monotherapy. This section describes the initial trials designed after failure of VEGF-TKI monotherapy and subsequent sections describe the more contemporary trials in the era of post-TKI/IO or IO/IO therapy.

### 3.1. Second-Line Therapies after VEGF Monotherapy

Efficacy of second-line therapies after failure of first-line VEGF-TKI monotherapy was limited and initial drug approvals of second-line agents were mainly investigated after previous treatment with VEGF-TKI monotherapy, with available evidence suggesting use of VEGF-TKI therapy or immunotherapy over mechanistic target of rapamycin (mTOR; formerly known as mammalian target of rapamycin) inhibitor therapy as the preferable second-line option. The typical sequencing options included using drugs that have a different mechanism of action that were previously used as first-line treatment. Therefore, the first mTOR-targeted pathway drug that was approved by the FDA for advanced RCC was temsirolimus, based on an initial trial that compared temsirolimus to interferon-alpha (IFN-$\alpha$) and the combination of IFN-$\alpha$ and temsirolimus in a 1:1:1 randomization, with the primary endpoint in a predominantly poor-risk patient population of OS, in an ITT analysis, that was achieved and ultimately brought about the FDA approval in 2007 [17], although comparison of temsirolimus to sorafenib showed no statistically significant difference in the treatment arms for PFS [18].

However, when compared to sorafenib, there was no statistically significant difference in PFS between the treatment arms.

There was evidence to suggest that switching from one VEGF-TKI to another does have utility and benefit. Therefore, the AXIS trial, a phase III registration study comparing axitinib to sorafenib as a second-line therapy, evaluated PFS as the primary endpoint and demonstrated a statistically significant benefit in favor of axitinib with a PFS of 8.3 months (95% CI 6.7–9.2) compared to 5.7 months with sorafenib (HR 0.656, 95% CI 0.552–0.779; one-sided $p < 0.0001$) [19]. Although the trial did not show a difference in OS between the two arms (median OS of 20.1 months with axitinib and 19.2 months with sorafenib, HR 0.969, 95% CI 0.800–1.174; one-sided $p = 0.3744$), it was not the primary endpoint of the trial, which was PFS. Regardless, this trial's results ultimately led to approval by the US FDA of axitinib in 2012 after failure of prior first-line therapy typically including an anti-angiogenic agent.

The utilization of immune checkpoint inhibitors as second-line therapy for metastatic RCC was initially approved by the FDA through the CheckMate 025 trial [20], which was an open-label, phase III registrational randomized trial, with a primary endpoint of OS. The trial demonstrated a median OS of 25.0 months for nivolumab, compared to 19.6 months for everolimus, with an HR of 0.73; 98.5% CI, 0.57–0.93, $p = 0.002$ for those with prior VEGF-TKI therapy. This trial led to the FDA's first approval of nivolumab for previously treated advanced RCC after anti-angiogenic therapy in 2015.

Another VEGF-TKI targeted switch approach came in the form of cabozantinib, a VEGF inhibitor against multiple targets including RET, AXL, MET, FLT3, VEGFR2, and c-KIT, especially since mounting evidence suggested targeting AXL and MET could overcome resistance to sunitinib, leading to the initiation of the phase III registrational randomized METEOR trial in the same population of previously treated patients on VEGF-inhibitors [21]. The METEOR trial compared cabozantinib with everolimus with a primary endpoint of PFS and resulted in a median PFS of 3.8 months in the everolimus arm compared to 7.4 months in the cabozantinib arm, which led to the US FDA approval of cabozantinib in 2016 for advanced RCC patients who were previously treated with anti-angiogenic agents [21].

In a randomized multicenter trial, a combination of lenvatinib and everolimus was evaluated in a smaller phase II study. The trial compared 18 mg of lenvatinib and 5 mg of everolimus to 24 mg of lenvatinib monotherapy and 10 mg of everolimus alone in a 1:1:1 fashion, with the investigator-assessed PFS as the primary endpoint. The results showed that the lenvatinib plus everolimus arm had a median PFS of 14.6 months (95% CI: 5.9–20.1), which was significantly better than the 5.5 months (95% CI: 3.5–7.1) for patients in the everolimus arm. Based on these positive results, the US FDA approved the combination of lenvatinib and everolimus as a second-line treatment option after prior treatment in 2016 [22].

The most recent small-molecule, potent, and selective VEGF inhibitor studied and US FDA-approved in 2021 was tivozanib, based on the Tivo-3 trial [23], which established the safety and efficacy of tivozanib. In this trial, 1.5 mg of tivozanib given in 4-week cycles was compared to sorafenib 400 mg twice daily. The primary endpoint was PFS. The results showed an improvement in median PFS for tivozanib at 5.6 months compared to sorafenib at 3.9 months (HR 0.73, 95% CI 0.56–0.94; $p = 0.016$) for patients with advanced RCC who had experienced two or more prior systemic therapy failures. It is worth noting that the patient population in this trial had been heavily pre-treated with multiple different pathways of therapy. Additionally, tivozanib has demonstrated a better tolerability and safety profile compared to sorafenib.

### 3.2. Second-Line Treatment after Contemporary First-Line IO/IO or IO/VEGF-TKI Regimens

The second-line space in the management of advanced clear cell renal cell carcinoma has certainly become more complicated in recent years with the advancements made in the first-line setting. To date, prospective trial data on second-line trials which enrolled patients who previously received contemporary first-line regimens are limited [24], with little guidance on appropriate sequencing strategies. However, the trials can be generally divided into those that examine further TKIs after prior IO/VEGF-TKI regimens or use of IO regimens as salvage therapy or further intensification after prior failure (see Table 2).

**Table 2.** Select key studies for second-line treatment with metastatic RCC.

| NCT# (Phase) | N | Title | Primary Endpoint(s) | Arms | Results |
|---|---|---|---|---|---|
| NCT01136733 (Ph 2) | 153 | Lenvatinib, Everolimus, and the combination in patients with metastatic renal cell carcinoma. | PFS | Lenvatinib + everolimus (*n* = 51) vs. lenvatinib (*n* = 52) vs. everolimus (*n* = 50) | 14.6 months vs. 7.4 months vs. 5.5 months |
| NCT02579811 (Ph 2) | 40 | Individualized axitinib regimen for patients with metastatic renal cell carcinoma after treatment with checkpoint inhibitors. | PFS | Single arm axitinib | 8.8 months |
| NCT03203473 (Ph 2) | 83 | Optimized management of nivolumab and ipilimumab advanced renal cell carcinoma: Response based phase II study. | PR/CR | Nivolumab induction (arm A) and ipilimumab conversion (arm B) | ORR of induction nivolumab = 12% (arm A) and PR 14% (arm B) |
| NCT020501096 (Ph 1b/2) | 143 | Lenvatinib plus pembrolizumab in patients with either treatment naïve or previously treated metastatic renal cell carcinoma (study111/keynote 146): Phase 1B/2 study. | ORR at 24 weeks | Lenvatinib plus pembrolizumab | 16/22 (72.7%) treatment-naïve 7/17 (41.2%) ICI naive58/104 (55.8%) ICI pre-treated |

**Table 2.** *Cont.*

| NCT# (Phase) | N | Title | Primary Endpoint(s) | Arms | Results |
|---|---|---|---|---|---|
| NCT04987203 (Ph 3) | Ongoing recruitment (326 planned) | TiNivo-2: A phase 3 randomized controlled multicenter open label study to compare tivozanib in combination with nivolumab to tivozanib monotherapy in subjects with renal cell carcinoma who have progressed following one or two lines of therapy where one line had immune checkpoint inhibitor. | PFS | Tivozanib + nivolumab vs. tivozanib monotherapy | Awaited |
| NCT04338269 (Ph 3) | 523 | Contact-03: Randomized, open label phase 3 study of atezolizumab plus cabozantinib versus cabozantinib monotherapy following progression on/after immune checkpoint inhibitor treatment in patients with advanced/metastatic renal cell carcinoma. | PFS OS | Atezolizumab + cabozantinib versus cabozantinib monotherapy | Press release showed it did not meet primary endpoint of PFS |
| NCT00678392 (Ph 3) | 723 | Axitinib versus sorafenib as second line treatment for advanced renal cell carcinoma. | PFS | Axitinib versus sorafenib | 8.3 months versus 5.7 months |
| NCT01668784 (Ph 3) | 821 | Nivolumab versus Everolimus in Advanced Renal-Cell Carcinoma. | OS | Nivolumab versus everolimus | 25.0 months versus 19.6 months |
| NCT01865747 (Ph 3) | 658 | Cabozantinib versus Everolimus in Advanced Renal-Cell Carcinoma. | PFS | Cabozantinib versus everolimus | 7.4 months versus 3.8 months |
| NCT02627963 (Ph 3) | 350 | Tivozanib versus sorafenib in patients with advanced renal cell carcinoma (TIVO-3): a phase 3, multicentre, randomised, controlled, open-label study. | PFS | Tivozanib versus sorafenib | 5.6 months versus 3.9 months |

PFS = progression-free survival; OS = overall survival; ORR = objective response rate.

In 2015, a prospective trial was conducted on second-line lenvatinib and everolimus (each as monotherapy and in combination) [22] which showed a PFS benefit in patients with one prior line of VEGF-targeted therapy. However, only five out of 153 patients had

also received prior IO therapy, indicating that the benefit of these regimens could not be assumed in the post-IO setting [22]. Nonetheless, a retrospective analysis conducted in 2021 demonstrated the efficacy of lenvatinib with or without everolimus in patients previously treated with both IO and VEGFR-targeted therapies. The study involved 55 patients with a median of four prior therapies, all of whom had received both an immune checkpoint inhibitor and a VEGFR-TKI. The majority of patients had clear cell histology, and 18% had non-clear cell RCC. Among all patients treated with lenvatinib with or without everolimus, the ORR was 21.8%, with one patient achieving a CR and 11 achieving a PR. The median OS was 12.1 months [25]. In 2019 and 2020, several additional retrospective studies were published lending further support regarding the efficacy of VEGF-targeted therapies after progression of disease on an immune checkpoint inhibitor regimen that was administered in the first-line setting [26–28].

In a phase II, prospective study reported by Ornstein and colleagues in 2019, axitinib was administered to 40 patients who had received a checkpoint inhibitor as part of their most recent preceding treatment. The study allowed an individualized dosing algorithm, and there were no restrictions on the number of prior treatments received. However, it is worth noting that 27% (11 out of 40) of patients had only received one prior line of therapy, and the majority of patients (70%) had previously received VEGF-targeted therapy. Most of these patients had received nivolumab monotherapy, and a smaller proportion (15%) had received combination ipilimumab and nivolumab as their preceding checkpoint inhibitor regimen. Although the trial did not meet its primary endpoint, at a median follow-up of 8.7 months, the median PFS for all patients was 8.8 months and the ORR was 45%. Among the responders, one patient achieved a CR, and 17 patients had a PR, while 18 patients had stable disease (SD) as their best response. Axitinib resulted in a sustained response of over 12 months in 67% of the responding patients. In a post hoc subgroup analysis, the median PFS in patients who stopped prior checkpoint inhibitor therapy for disease progression was 9.2 months. Among the eleven patients who received axitinib on trial as a second-line therapy, four (36%) had an objective response to axitinib [29].

OMNIVORE was a phase II trial that aimed to evaluate the efficacy of IO therapy in metastatic RCC patients who had not previously received IO treatment [30]. Eighty-three patients were enrolled and treated with nivolumab monotherapy, with subsequent treatment allocation based on their response to nivolumab. Patients with a PR or CR within six months of treatment cessation were placed on observation, while those with stable (SD) or progressive disease (PD) after no more than six months of nivolumab received two doses of ipilimumab to convert and salvage responses. The primary endpoints of the trial were the proportion of patients with PR/CR at one year after nivolumab discontinuation (arm A) and the proportion of nivolumab non-responders who were converted to PR/CR after receiving ipilimumab (arm B). Of the 83 patients, 14 did not undergo arm allocation due to toxicity or disease progression. Of the remaining 69 patients, 12 were allocated to arm A (10 PR, 1 unconfirmed PR, and 1 stable disease) and 57 were assigned to arm B (28 SD, 29 PD). The ORR of induction nivolumab was 14%, with a higher rate of 17% in treatment-naïve patients and a lower rate of 12% in previously treated patients. Of the twelve patients on observation in arm A, five remained off nivolumab at one year post-treatment discontinuation. Four patients resumed nivolumab within 6 months of discontinuation, with three due to PD and one still in PR. Three additional patients had not reached the 1-year mark and were still under follow-up at the time of publication. Of the fifty-seven patients assigned to arm B, two patients with PD on nivolumab converted to a confirmed PR on ipilimumab, with a duration of response of 9.2 and 10.9 months, respectively, for each of these patients after the addition of ipilimumab. The best response in arm B patients was stable disease in 46% and progressive disease in 40%, with a median duration of disease control of 10.4 months. The median follow-up for OS was 19.5 months, and the median OS was not reached, with an 18-month overall survival rate of 79%. However, due to the small number of patients allocated to arm A and the low rate of conversion to response in arm B, the trial did not recommend a response-adaptive approach to IO therapy based on its findings.

Similar to OMNIVORE, TITAN-RCC was also a response-adaptive study of IO therapy in the first- (*n* = 109) and second-line (*n* = 98) setting for patients with intermediate- and poor-risk RCC [31], with final analyses reported at the 2022 ESMO Congress [32]. In this trial, patients received nivolumab induction which entailed dosing every 2 weeks for a total of eight doses. Those who achieved a CR or PR on nivolumab continued on nivolumab maintenance until progression of disease and those whose best response was SD or PD received up to two sets of boost therapy with combination ipilimumab plus nivolumab given twice three weeks apart. If CR/PR was achieved after one boost, patients transitioned to nivolumab maintenance. Those with persistent SD/PD after one boost received an additional boost after which they transitioned to nivolumab maintenance if CR, PR, or SR was achieved. A confirmed response to nivolumab induction was seen in 28% and 18% of first-line and second-line patients, respectively. Forty-four percent of first-line patients who received induction nivolumab followed by ipilimumab plus nivolumab boosts for PD had an improvement in best response. However, 53% of second-line patients who received induction nivolumab followed by ipilimumab plus nivolumab boosts for PD had similar improvements in response. OS was 32 months and 25.9 months in first- and second-line patients, respectively. While these reported outcomes are inferior to those demonstrated by upfront dual checkpoint inhibition with nivolumab and ipilimumab, they do support the addition of ipilimumab for more vulnerable patients in whom nivolumab is used alone initially to minimize toxicity, but the response to monotherapy is suboptimal [32].

KEYNOTE-146 was a phase Ib/II open-label study assessing the efficacy of lenvatinib in combination with pembrolizumab. One hundred and four out of one hundred and forty-three metastatic clear cell RCC patients were previously treated with an immune checkpoint inhibitor. ORR at week 24 was 55.8% for ICI-pre-treated patients. All responses were partial responses. Among responders, the duration of response was 12.5 months in these patients. At a median follow-up of 18.9 months, median PFS was 12.2 months for those previously treated with an immune checkpoint inhibitor [33].

The TiNivo-2 study is a randomized, controlled, open-label phase III trial that aims to compare the efficacy of tivozanib plus nivolumab with tivozanib monotherapy in patients with advanced RCC who have progressed on one or more lines of treatment and were previously treated with an immune checkpoint inhibitor [34]. This trial is now open for enrollment and will hopefully further address the question of whether continuation of an immune checkpoint inhibitor when combined with a new TKI provides any benefit compared to a TKI monotherapy regimen after progression of disease on IO therapy, which remains the current standard of care. CONTACT-03 is another phase III trial looking at a similar patient population of ICI-pre-treated patients and investigating the use of atezolizumab in combination with cabozantinib compared to cabozantinib monotherapy [35], although a recent press release indicated that the study did not meet its primary endpoint. Taken together, these two trials serve to answer the question of utility of continuation of the IO therapy backbone in a patient whose disease is progressing on IO therapy.

There are several clinical trials studying belzutifan, an oral HIF-2α inhibitor, in combination with other agents, such as palbociclib (NCT05468697), lenvatinib (NCT04586231), or cabozantinib and PT2977 (NCT03634540) (see Table 3), currently recruiting previously treated metastatic RCC patients who have received an immune checkpoint inhibitor. Moving forward, we hope to gain a better understanding of the role that novel therapeutic agents and combinations play in the second- and third-line setting in an era where the majority of advanced ccRCC patients receive a VEGR-TKI plus IO or dual IO regimen in the first-line setting.

**Table 3.** Key Studies of Belzutifan (MK-6482) in Advanced RCC.

| NCT# (Phase) | N | Description | Primary Endpoint(s) | Completion Date |
|---|---|---|---|---|
| NCT04586231 (Ph 3) | 708 | A study of belzutifan in combination with lenvatinib versus cabozantinib for treatment of renal cell carcinoma | PFS OS | 23 December 2024 |
| NCT03634540 (Ph 2) | 118 | A trial of belzutifan in combination with cabozantinib in patients with clear cell renal cell carcinoma | ORR | 31 August 2025 |
| NCT04195750 (Ph 3) | 736 | A study of belzutifan versus everolimus in participants with advanced renal cell carcinoma | PFS OS | 17 September 2025 |
| NCT04489771 (Ph 2) | 150 | A study of belzutifan in participants with advanced renal cell carcinoma | ORR | 4 October 2025 |
| NCT04736706 (Ph 3) | 1653 | A study of pembrolizumab in combination with belzutifan and lenvatinib or pembrolizumab/quavo in combination with lenvatinib, versus pembrolizumab and lenvatinib, for treatment of advanced clear cell renal cell carcinoma | PFS OS | 29 October 2026 |
| NCT05468697 (Ph 1/2) | 180 | A study of belzutifan in combination with palbociclib versus belzutifan monotherapy in participants with advanced renal cell carcinoma | ORR (Part 2) | 16 May 2027 |

PFS = progression-free survival; OS = overall survival; ORR = objective response rate.

## 4. Future Directions

As we look ahead, we anticipate the incorporation of belzutifan and other novel targeted agents into mRCC treatment regimens. For example, MK-6482-012 (NCT04736706) is a clinical trial investigating triplet regimens (pembrolizumab/belzutifan/lenvatinib and pembrolizumab/quavonlimab/lenvatinib) versus standard-of-care pembrolizumab plus lenvatinib in the first-line setting. The management of mRCC patients beyond first-line treatment has already become more complicated now that triplet therapy regimens have shown improvement in progression-free survival over current standard-of-care doublet regimens, although it remains to be seen how and which population they would be most utilized in. Similarly, the utility of switching over to regimens with different mechanisms of action continues to be an important clinical question. Additionally, the approval and use of adjuvant pembrolizumab in high-risk localized or oligometastatic RCC post-nephrectomy (including those rendered M0 post-metastatectomy) based on the KEYNOTE-564 trial add to the complexity of managing select patients who develop metastatic relapse after definitive local therapy followed by adjuvant ICIs. The benefits, if any, of continuing or resuming ICIs in these patients are unknown at present and this is an area of unmet need. In the meantime, we could anticipate extrapolating data from the aforementioned second-line trials that are addressing the use of ICIs beyond first-line treatment for those who have been previously treated with ICIs for de novo mRCC. Another thing to consider is the potential addition of novel agents to pembrolizumab in the adjuvant setting, which is currently being evaluated in the LITESPARK-022 trial, as one example. As further advancements are made in the adjuvant, post-nephrectomy space, the treatment of those who relapse during or after adjuvant systemic therapy will remain an area of unmet need. Surely, it is a good problem to have and we recognize the sincere efforts being made by the research community to keep up with the rapidly evolving care of mRCC patients in spite of the many questions and clinical dilemmas we will continue to encounter.

## 5. Conclusions and Recommendations

The field of mRCC treatment is continuously evolving. The choice of first-line therapy is dependent on multiple varying factors including patient characteristics, IMDC factors, the presence of critical visceral crisis, patient preference, and physician familiarity or formulation availability [36,37]. While certain combinations, particularly IO and TKIs, appear to result in better objective responses, dual CPI use with PD-1 and CTLA-4 inhibition perhaps leads to durable responses with the advantage of longer-term memory T cell function. However, toxicity profiles are vastly different. Physicians are now much more familiar with managing immune-related toxicities although it remains important to understand that availability and access to subspecialty resources are critical. In addition, the role of triplet therapy continues to evolve. While COSMIC 313 did meet its primary endpoint of PFS, the potential added magnitude of benefit appears to be small, and the increased toxicity would be worthwhile to pay close attention to. Other considerations include the impact of disease progression in patients who have received or are still undergoing adjuvant pembrolizumab therapy, which is the topic of multiple clinical trials. While patients who receive dual IO treatment as front-line therapy usually proceed to second-line VEGF-TKI with numerous contemporary trials supporting this, there is less consensus with regard to second-line therapy in those who have received IO/TKI options as front-line therapy. Continuation or a switch to a different IO agent is not currently standard of care but several trials are studying the utility of this approach.

**Author Contributions:** Conceptualization, J.B.A.-C.; methodology, J.B.A.-C., M.M., I.P., E.-m.Y.; software, J.B.A.-C., M.M., I.P., E.-m.Y.; formal analysis, J.B.A.-C., M.M., I.P., E.-m.Y.; investigation, J.B.A.-C., M.M., I.P., E.-m.Y.; data curation, J.B.A.-C., M.M., I.P., E.-m.Y.; writing—original draft preparation, J.B.A.-C., M.M., I.P., E.-m.Y.; writing—review and editing, J.B.A.-C., M.M., E.-m.Y.; supervision, J.B.A.-C. and E.-m.Y. All authors have read and agreed to the published version of the manuscript.

**Funding:** This research received no external funding.

**Institutional Review Board Statement:** Not applicable.

**Informed Consent Statement:** Not applicable.

**Data Availability Statement:** Not applicable.

**Conflicts of Interest:** JAC serves on the Speakers' Bureau of BMS, Astellas/SeaGen, Pfizer/EMD Serono and has served on the Advisory Board of Astellas/Seagen, Pfizer/EMD Serono, Merck, Janssen, Exelixis, AVEO, Bayer. All other authors declared that there are no conflict of interest.

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
