# Peer review of "Evolving Treatment Options for Metastatic Renal Cell Carcinoma (mRCC)"

_2673-4397, doi:10.3390/uro3020014_

Round 1

Reviewer 1 Report

Very good quality narrative review on the complex topic of treatment of mRCC.   

In-depth analysis on critical issues like:

- IO/VEGF-TKI combination therapies

- triple regiments, 

- further complication of second line options in the last few years  

The manuscript is well structured, with contemporary references and including all the breakthrough RCT on the subject according to this reviewer`s opinion.  

My recommendation is to accept this manuscript for publication in its present form.  

Author Response

Response to Reviewer: We highly appreciate the Reviewer’s recommendations and have made additional changes to the manuscript as per recommendations above.  Added in-depth analyses of different IO/TKI combination analyses, triplet regiments include COSMIC-313 was mentioned and we have a section on second-line options after contemporary first-line treatment strategies.

Reviewer 2 Report

The Authors propose a narrative review (even if not stated) on treatments for metastatic cell renal cancer. 

I believe that the work could be a good didactic tools for its huge of the treated arguments. 

The work is well written, its contents are effective and the methodology is correct. Overall, the manuscript is scientifically sound. 

Therefore, I would offer them some formal suggestions.

First of all, in the tables they are suggested to cite the study reported. 

In addition, I strongly suggest to offer a timeline or a full scheme of the evolving treatment strategies, showing the drugs and the molecular target.

I remain at your disposal.

Best Regards

Author Response

Response to Reviewer: Thank you very much to the Reviewer for this feedback.  We have cited the studies in the table using the clinicaltrials.gov NCT number identifiers especially for those that are not yet published.

We have added a timeline or schematic of the evolving treatment strategies and added a figure (Figure 1) as shown in the attachment.